# Artificial Intelligence Distinguishes Pathological Gait: The Analysis of Markerless Motion Capture Gait Data   Acquired by an iOS Application (TDPT-GT)

**DOI:** 10.3390/s23136217

**Published:** 2023-07-07

**Authors:** Chifumi Iseki, Tatsuya Hayasaka, Hyota Yanagawa, Yuta Komoriya, Toshiyuki Kondo, Masayuki Hoshi, Tadanori Fukami, Yoshiyuki Kobayashi, Shigeo Ueda, Kaneyuki Kawamae, Masatsune Ishikawa, Shigeki Yamada, Yukihiko Aoyagi, Yasuyuki Ohta

**Affiliations:** 1Division of Neurology and Clinical Neuroscience, Department of Internal Medicine III, Yamagata University School of Medicine, Yamagata 990-2331, Japan; toshikon@med.id.yamagata-u.ac.jp (T.K.); yasuyuki@med.id.yamagata-u.ac.jp (Y.O.); 2Department of Behavioral Neurology and Cognitive Neuroscience, Tohoku University Graduate School of Medicine, Sendai 980-8575, Japan; 3Department of Anesthesiology, Yamagata University School of Medicine, Yamagata 990-2331, Japan; hayasakatatsuya1101@gmail.com (T.H.); aequanimitas.3520@gmail.com (Y.K.); 4Department of Medicine, Yamagata University School of Medicine, Yamagata 990-2331, Japan; m181274@st.yamagata-u.ac.jp; 5Department of Physical Therapy, Fukushima Medical University School of Health Sciences, 10-6 Sakaemachi, Fukushima 960-8516, Japan; mhoshi@fmu.ac.jp; 6Department of Informatics, Faculty of Engineering, Yamagata University, Yonezawa 992-8510, Japan; fukami@yz.yamagata-u.ac.jp; 7Human Augmentation Research Center, National Institute of Advanced Industrial Science and Technology (AIST), Kashiwa II Campus, University of Tokyo, Kashiwa 277-0882, Japan; kobayashi-yoshiyuki@aist.go.jp; 8Shin-Aikai Spine Center, Katano Hospital, Katano 576-0043, Japan; uedashigeo@yahoo.co.jp; 9Department of Anesthesia and Critical Care Medicine, Ohta-Nishinouti Hospital, Koriyama 963-8558, Japan; kkawamae@ohta-hp.or.jp; 10Rakuwa Villa Ilios, Rakuwakai Healthcare System, Kyoto 607-8062, Japan; rakuwadr1001@rakuwadr.com; 11Normal Pressure Hydrocephalus Center, Rakuwakai Otowa Hospital, Kyoto 607-8062, Japan; shigekiyamada393@gmail.com; 12Department of Neurosurgery, Nagoya City University Graduate School of Medical Science, Nagoya 467-8601, Japan; 13Interfaculty Initiative in Information Studies, Institute of Industrial Science, The University of Tokyo, Tokyo 113-8654, Japan; 14Digital Standard Co., Ltd., Osaka 536-0013, Japan; y.aoyagi@digital-standard.com

**Keywords:** artificial intelligence, motion tracking, markerless motion capture, quantitative gait assessment, smartphone device, neuromuscular diseases

## Abstract

Distinguishing pathological gait is challenging in neurology because of the difficulty of capturing total body movement and its analysis. We aimed to obtain a convenient recording with an iPhone and establish an algorithm based on deep learning. From May 2021 to November 2022 at Yamagata University Hospital, Shiga University, and Takahata Town, patients with idiopathic normal pressure hydrocephalus (*n* = 48), Parkinson’s disease (*n* = 21), and other neuromuscular diseases (*n* = 45) comprised the pathological gait group (*n* = 114), and the control group consisted of 160 healthy volunteers. iPhone application TDPT-GT captured the subjects walking in a circular path of about 1 meter in diameter, a markerless motion capture system, with an iPhone camera, which generated the three-axis 30 frames per second (fps) relative coordinates of 27 body points. A light gradient boosting machine (Light GBM) with stratified k-fold cross-validation (k = 5) was applied for gait collection for about 1 min per person. The median ability model tested 200 frames of each person’s data for its distinction capability, which resulted in the area under a curve of 0.719. The pathological gait captured by the iPhone could be distinguished by artificial intelligence.

## 1. Introduction

In clinical settings, pathological gait must be distinguished for recognizing symptoms, diagnosis/differential diagnosis, and monitoring disease progression and the effects of treatment and rehabilitation. Gait evaluation is complicated because it includes many factors such as speed, power, position, angle, and cycle on each body part, expressing posture, arm and leg swings, and steps. Furthermore, pathological gait is presented in many combinations: asymmetry, unbalance, defragmentation, and reduced speed, length, amplitude, and smoothness. Recognizing, diagnosing, and even describing all these symptomatic disturbances are always challenging.

To evaluate gait, various quantified methods have been employed: pedobarography, motion capture, floor sensors, and wearable sensors. Motion capture is useful for understanding whole-body movements, while others usually concentrate on parts of the body. Optical motion capture systems require the attachment of several markers to the body, usually with multiple cameras [1,2,3,4,5,6,7,8]. Marker-based motion capture for gait requires the attachment of markers from head to toe during preparation and when recording for a long time. Older people and patients, sometimes including individuals with cognitive impairment, may not tolerate that. Therefore, most marker-based studies have focused on healthy volunteers or athletes.

Active markerless motion capture systems emit light [9,10,11,12] or micro-doppler [13] to collect the structures of objects, which requires a prepared place and settings like a laboratory. Passive systems rely on only the images captured by cameras and take a relatively long time for analysis to generate the gait data after the recording [14,15]. By attaching accelerometers and gyroscopes, called wearable devices, gait has been analyzed [7,16,17,18]. Some wearable devices are small and easy to wear; however, the information on gait is limited to speed and acceleration. Multimodal sensors and analysis systems with machine learning have been developed [10]; however, to date, the systems are not made for the convenience of participants but for obtaining precise information on gait.

Along with the history of gait analysis, especially for use in patients and older individuals, clinicians need non-invasive markerless motion capture systems that are possible to use in the general room for outpatients and that take participants as short a time as possible. In this study, we used a passive markerless motion picture system, TDPT-GT (Three-Dimensional Pose Tracker for Gait Test), which was novel in the circumstance in that the gait recording was just like taking videos on an iPhone [19,20]. In addition to the convenience and comfort for patients and older people, the system could immediately generate and preserve three-dimensional coordinate data of 27 body points.

Gait recognition techniques based on machine learning have been evolving [7], as well as our motion capture system [19]. Machine learning techniques have made it possible to collect complicated movements of human gait quickly. Then, the next challenge was how to analyze the large amount of gait data generated by these technologies. Machine learning has also been applied to the analysis of the gait data of neurological disorders [21,22,23]. Here, in this study, having acquired gait data in 30 frames per second (fps) for 27 body points with three axes produced by this iPhone application, we aimed to make artificial intelligence distinguish the motions of a pathological gait from those of a healthy gait.

## 2. Materials and Methods

### 2.1. Subjects

We collected information on age, disease history, and gait data from May 2021 to November 2022 at Yamagata University Hospital, Shiga University, and Takahata Town. The pathological gait group (*n* = 114) was composed of patients with idiopathic normal pressure hydrocephalus (iNPH, *n* = 48), Parkinson’s disease (PD, *n* = 21), and other neuro-muscular diseases (*n* = 55), which were mainly neurodegenerative diseases. The control group consisted of 160 individuals who were healthy volunteers at local health check-ups or family members of patients who did not have a neurodegenerative disease. The inclusion criterion for both groups was the capability of walking independently and safely for several minutes; using a single-point cane was the only assistance allowed.

### 2.2. Gait Data

The gait of each participant was recorded as they walked in a circular path of around 1 m in diameter. The recording was done from a distance of approximately 3 m away from the trail, ensuring it fit within the frame of the application. Each participant completed two laps, moving both clockwise and counterclockwise, as shown in the accompanying picture (Figure 1). They were asked to walk at their comfortable speed clockwise for two laps and counterclockwise for two laps. The application TDPT-GT (The versions were 20210525, 20211111, 20220314, 20220701, and 20220902) (Three-Dimensional Pose Tracker for Gait Test) is a markerless motion capture system based on machine learning. From the two-dimensional images taken by an iPhone camera, 3D heat maps of the body points were estimated. As shown previously [19,20], the TDPT-GT generated the three-axis 30 frames per second (fps) relative coordinates of 24 body points: the nose, navel, and bilateral points such as the eyes, ears, shoulders, elbows, wrists, thumbs, middle fingers, hip, knees, heels, and toes and calculated coordinates of three body points: the center of the head, neck, and buttocks. These coordinates were processed by a low-pass filter, named the 1 euro filter [24], setting the minimum cut-off frequency of 1.2, the cut-off slope of 0.001, and the cut-off frequency of the derivate of 1, and preserved as CSV files. We defined the navel as the reference point of the depth to each point. The raw X and Y coordinates were used to represent the vertical and horizontal points, respectively.

### 2.3. Deep Learning for the Distinction of Gait

About two hundred frames were extracted from the CSV files of each participant in an early and stable state of the record. All coordinates were labeled either disease gait or control gait. The total data were divided into 60 (*n* = 71–73 for disease and *n* = 102–104 for control) for training and 20 (*n* = 18–20 for pathological gaits and *n* = 24–26 for controls) for validation without dividing the data of the same participant into both datasets. (Table 1).

For deep learning, a light gradient boosting machine (Light GBM) [25] was employed, which was a technique involved combining a gradient boosting decision tree with gradient-based one-side sampling and exclusive feature bundling. The learning aimed to predict whether the gait was pathological in each frame containing the parameters of 27 three-dimensional body points. The hyperparameters were tuned by Optuna [26]; the method automatically selected the features and parameters that were determined to perform best. The feature importance scores of the coefficients from linear regression were calculated to identify the key points for predicting pathological gait. Stratified k-fold cross-validation (k = 5) was used to evaluate the differentiation ability of the models of machine learning by generating the classification accuracy, sensitivity, specificity, and area under the curve (AUC) calculated from the receiver operating characteristic (ROC) curve.

### 2.4. Test

To test the ability of machine learning, we used the newest (data acquisition order) 20 data out of the data of all patients and controls. Among the five models, the model with the median of the AUCs was employed for the final test to evaluate the model’s generalization. The final test was evaluated as follows to evaluate the gait of a whole person. The AUC was calculated from the ROC curve drawn based on the averaged accuracy for 200 frames for each person, and the sensitivity and specificity were also calculated from the curve. To optimize the hyperparameters, Optuna v.3.0 was used. The feature importance scores of the coefficients from the linear regression were calculated to identify the key points for predicting pathological gait.

### 2.5. Hardware, Software, and Statistics

Machine learning was carried out by Anaconda 3 on Ubuntu 20.04 LTS. The analysis hardware comprised a Core i9 10940X 14core CPU (Santa Clara, CA, USA) and an NVIDIA RTX A4000 16GB GPU (Santa Clara, CA, USA). EZR version 1.41 and Python 3.8 were used for the statistical analysis. The results were expressed as the mean ± standard deviation and numbers (percentages). The constructed model was considered to have sufficient diagnostic capability when AUC > 0.700, along with a 95% confidence interval (CI).

### 2.6. Ethical Considerations

The study was conducted according to the Declaration of Helsinki and approved by the Ethics Committee for Human Research of Shiga University of Medical Science (protocol code: R2019-337; date of approval: 17 April 2020) and the Ethical Committee of Yamagata University School of Medicine (protocol code: 2020-10; date of approval: 12 April 2021).

## 3. Results

### 3.1. Clinical Characteristics

The average age and standard deviation were 74.5 ± 7.8 years in the pathological gait group and 72.9 ± 11.1 years for the controls, which were not significantly different between the groups. Sex proportions were also not significantly different between the groups (Table 2).

### 3.2. Results of the Five Learning Models

Among the five models made by AI using stratified K-fold cross-validation (k = 5), dataset 1 presented the median of the AUCs: an AUC of 0.882 (the 95% confidence interval ranged from 0.875 to 0.890), a sensitivity of 0.740, a specificity of 0.898, and an accuracy of 0.8327 (Table 3, Figure 2).

Each learning model’s cut-off value (specificity and sensitivity) is shown near the curve.

#### 3.2.1. Discrimination of Pathological Gait

Machine learning model 1 made by dataset 1 was tested by the test data (Table 2). The result of distinguishing pathological gait was a cut-off of 0.595, an AUC of 0.719 (the 95% confidence interval was ranged from 0.576 to 0.862), a sensitivity of 0.652, a specificity of 0.781, and an accuracy of 0.709 (Figure 3).

#### 3.2.2. Feature Importance

The feature importance score was highest in the y coordinate of the right hip joint, followed by high scores in the depth of the buttock, the depth of the right knee, and the x coordinate of the center of the head (Figure 4).

## 4. Discussion

In this study, AI distinguished the pathological gait from the control gait with an AUC of 0.719 for the ROC and an accuracy of 0.702. The markerless motion capture system on the iPhone successfully recorded the gaits for analysis. The present analysis included two valuable methods. First, the motion capture of gaits was obtained very easily with an iPhone camera and its application TDPT-GT. The participants did not need to wear markers or devices when walking in a circle within the camera’s angle of view for just several minutes at a comfortable speed. Second, we succeeded in diagnosing pathological gait by constructing an AI analysis for the big and precise data from TDPT-GT, with 30 fps and 27 body points in 3 dimensions.

### 4.1. Usage of the Present AI Model

The present AI model has been trained to effectively differentiate pathological gait, a task that was challenging even for experts. Clinicians do not solely rely on visual observation of a patient’s gait to determine if it is pathological. In the early stages of disease, patients typically do not exhibit apparent gait disturbances when they come for a visit. Therefore, we collect many clinical records, such as backgrounds, history, labo data, systematic neurological examinations, physiological examinations, and neuroimaging. Subsequently, other diseases are also differentiated and/or judged as complications. Gait provides just a piece of information. After this process, we diagnose the disease and comprehensively ensure that the person’s gait is pathological. AI distinguished the gait without these long, clinical, and complicated procedures. Consequently, AI’s incorporation into gait screening during health check-ups in communities of older individuals can lead to earlier diagnoses.

Our ultimate objective in the future is not only to achieve the early diagnosis of pathological gait but to also promote overall well-being. The ability to ambulate independently is a major contributor to overall well-being and autonomy in older individuals, and gait and its decline are crucial for the health and function of older patients [27]. Elderly patients regularly present with complex gait disorders, with concurrent contributions from multiple causal factors [27], such as muscle weakness, cognitive impairment, alcohol consumption, pain, and physical inactivity [28]. Among older individuals participating in a study, gait abnormalities were observed in 48.4% [29]. The combination of multiple modes of gait abnormalities including the self-reported gait difficulty predicted the risk of various geriatric outcomes, e.g., falls. [29]. Based on their study, we derived the notion that objectively detecting gait abnormalities should be carried out when individuals subjectively perceive difficulties in their gait. The ability of the present AI model to distinguish common pathological gait patterns is applicable for analyzing complex gait disorders arising from diverse causes. Our system aims to aid in identifying subjective gait disorders in order to prevent their progression.

Gait encompasses more than just the physical manifestation of movement. Jason reviewed and noted that the control of gait occurs via multiple cognitive domains [27]. Executive functions are a general system working with attention, memory, reasoning, and cognitive integration, which are associated with gait parameters, such as velocity and step length [30,31,32]. Gait velocity is related to cognitive processing speed [30], short-term memory [30], and multiple cognitive domains [33]. By capturing gait disturbances, we can also gain insights into recognizing cognitive decline and may contribute to overall well-being.

The discrimination ability in the study was an accuracy of 0.709, with a sensitivity of 0.652 and a specificity of 0.781. di Biase introduced other studies that tried to discriminate the gait of Parkinson’s disease by various measurements from control subjects with at least 80% sensitivity and specificity [21]. Most previous studies have included clinically engaged, diagnosed patients and analyzed gait through complicated monitoring [34,35,36,37,38]. We supposed that the gait impairment of Parkinson’s disease was one of the most characteristic among other diseases and not difficult to diagnose with other symptoms without gait information. Hence, it was possible to be promising to discriminate among their participants. The discriminatory ability in our study was diminished due to the inclusion of various diseases. Recognizing the challenges that arose from the hardships involved in discriminating among our study subjects, we undertook the challenge.

With advancements in AI and the motion capture technology of TDPT, there is potential for further development to differentiate or indicate various types of gait disturbances. Gait movement disorders are expressed as wide-based, shuffling, dragging, small steps, hesitating, frozen, propulsive, waddling, swaying, spastic, and ataxic. Even among specialists, certain gait symptoms exhibit limited inter-rater reliability [39]. In addition, patients’ symptoms fluctuate naturally, affected by circumstances, timing, and fatigue. These aspects lie beyond our reach during the examinations conducted at the hospital. The system implemented in the study will assist us in perceiving and analyzing diverse gait disorders in various settings.

### 4.2. Gait Analysis by TDPT-GT

TDPT-GT is an application of motion capture for various possible utilities, such as making animations of humans and avatars and analyzing posture, dance, and sports. For human health, the application may increase knowledge of human motion and may help diagnose symptoms. It is significant that this application works on the iPhone with the precise camera characteristics of the device and automatically generates the data of many points of the entire body, including the head, trunk, arms, and legs. Furthermore, the data incorporate the notion of time.

AI studies of movement in Parkinson’s disease (PD) were reviewed by Biase [21], who focused on several features, for example, the maximum angles of a joint, stride length, swings, and their statistical indexes. Each study tried to find characteristic disease features among them. Hereditary spastic paraplegia and cerebral palsy were discriminated by AI analysis of measures including strength and spasticity and kinematic gait metrics, totaling 179 variables [22]; however, those measures were impracticable in clinical settings. A smartphone was used to collect the tri-axial acceleration, and secondary associated statistical data were used to discriminate between PD and controls through AI analysis [23]. The advantage of our study was that the gait data were all primarily obtained automatically by a smartphone.

The present study’s algorithm of distinction of gait was learned frame by frame, moment by moment, and one by one. Additionally, the pathological gaits were distinguished from those of the controls by the entire gait time of a person. Although the learning method was simple, gait was evaluated with the concept of time because the whole gait of the subjects was analyzed for around one minute. Compared with previous studies of gait analysis with AI, we employed the most convenient and low-cost motion capture, using Al analysis but systematically for the whole human body and the gait process.

#### Discoveries Made by AI

Feature importance, which indicates particular coordinates that strongly affect the decision of AI analysis, is being used in many fields of medicine, such as genetics [40], the drug design field [41], and survival prediction [42]. In the study, high feature importance scores were found in the coordinates of the hip joint, the buttocks, the knee, and the center of the head. Despite the inherent lack of interpretability in AI analysis, it was observed that the findings predominantly pertained to the body’s trunk. We usually describe the gait mainly as the movement of the legs; however, the model seemed to focus on posture during walking. Therefore, these results of feature importance suggested that we should reconsider the clinical importance of posture and the trunk while walking. Such findings may be valuable in reconstructing medical assessments and rehabilitation knowledge. The results of the present study implied that pathological gaits, mainly with hydrocephalus and neurodegenerative diseases, had common mechanisms of posture impairment. Static posture and posture during gait are also objected to analysis with sensors [10,43,44,45,46,47]. In neurodegenerative diseases, Parkinson’s disease is the most analyzed for its posture impairment [10,48,49]. In patients with Huntington’s disease, the wearable device revealed that the magnitude of thoracic and pelvic trunk movements was significantly higher during static periods (such as sitting or standing) compared to walking [46]. The function of the trunk is associated with balance and falls, expressing frailty, among older people without causative diseases [50]. Trunk posture adaptation before the onset of decline in gait speed was found in at-risk community-dwelling older adults [51]. Leveraging AI to detect trunk function will contribute to the early prevention and diagnosis of frailty or diseases. 

We suppose it is advisable not to separate the evaluation and discussion of posture and gait. For postural neural control, the mesencephalic area, the reticular formation, the forebrain structure, and the spinal locomotor networks, cerebral, cerebellar, and basal ganglia were affected [52]. The instability of the trunk can be attributed to various factors, such as peripheral nervous system disorders and psychosomatic disorders [53]. Trunk muscles are also crucial and responsive to training for the preventing falls [54]. Due to the complexity of the system, clinicians face challenges in recognizing disruptions in trunk function through direct observation. Our findings could facilitate earlier diagnosis or intervention for persistent impairments in posture and gait.

### 4.3. Limitations

The methodological limitation was that we were within the capability of the present smartphone (iPhone). The data we obtained from the system were 30 fps, while the VICON (Vicon Motion Systems, Oxford, United Kingdom), the standard for motion capture, could generate data of 100 Hz. More frequent analysis on the iPhone may exceed its processing ability and make the iPhone slower in its performance. In an effort to achieve convenience and versatility, as alternative motion capture systems, Microsoft Kinect [10,55] operated at a frame rate of 30 fps, while the Kinect of YOLOv4 [56] had a frame rate of 17.5 fps. Clinically, 30 fps seemed enough for human gait, which did not have a high-frequency motion, especially since patients needed to walk safely within the angle view of the iPhone camera. Even with the limitations of using an iPhone, we were able to swiftly obtain gait data directly at the hospital or the common space.

The clinical limitation of the study was that the severity of the pathological gait was difficult to describe and control. Therefore, learning may be easy if very severe pathological gaits are included. However, the patients with severe gait disturbance were not included naturally because the method of the gait test in the study needed the participants to walk safely and independently without assistance.

## 5. Conclusions

Gait data in the study were collected using a markerless motion capture system called TDPT-GT, utilizing an iPhone. The study involved examining patients with iNPH, PD, other neurodegenerative and neuromuscular diseases, and healthy volunteers. The deep learning algorithm could distinguish a pathological gait from a control gait with an AUC of 0.719 for the ROC and an accuracy of 0.702.

## Figures and Tables

**Figure 1 sensors-23-06217-f001:**
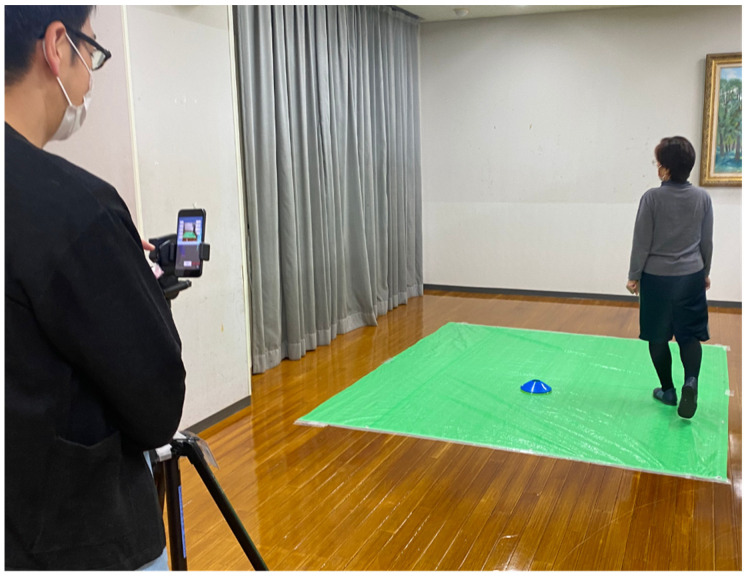
The markerless motion capture was recorded while the participants walked in a circle about 1 m in diameter and about 3 m away from the gait trail so that their whole body fit within the frame.

**Figure 2 sensors-23-06217-f002:**
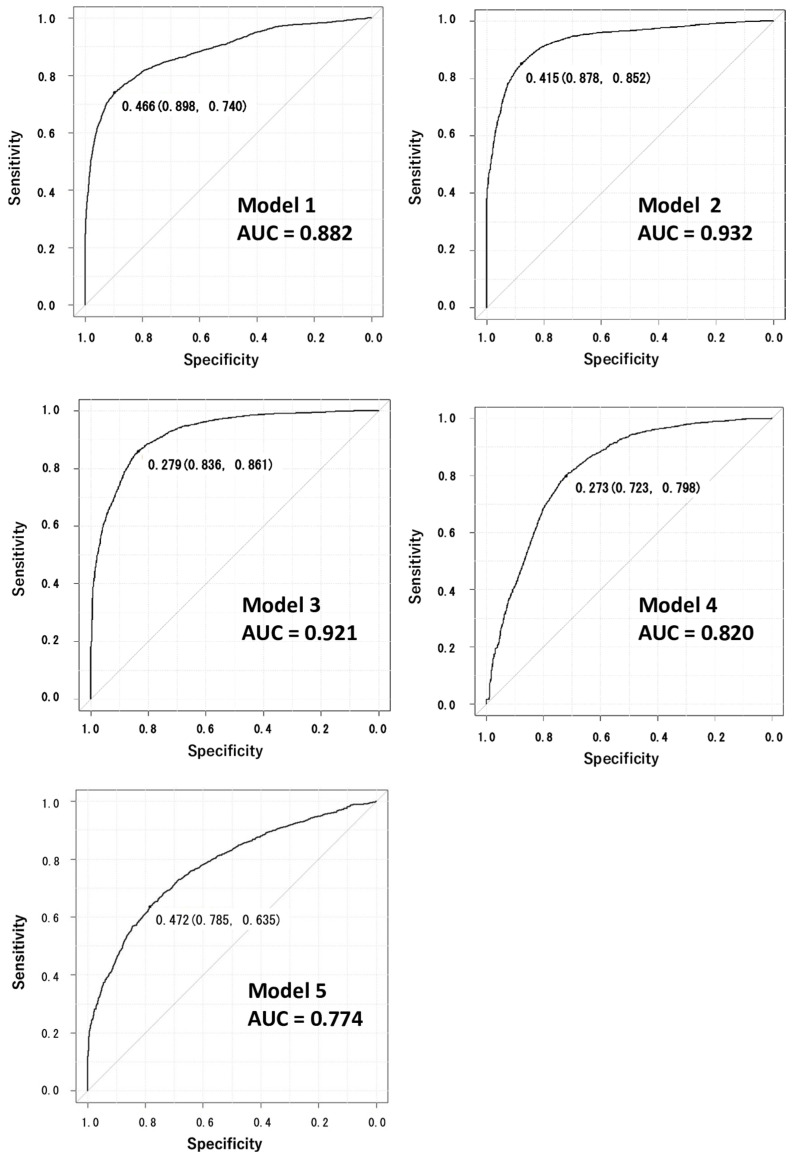
The receiver operating characteristic (ROC) curves and the area under the curve (AUC) obtained by the five machine learning models.

**Figure 3 sensors-23-06217-f003:**
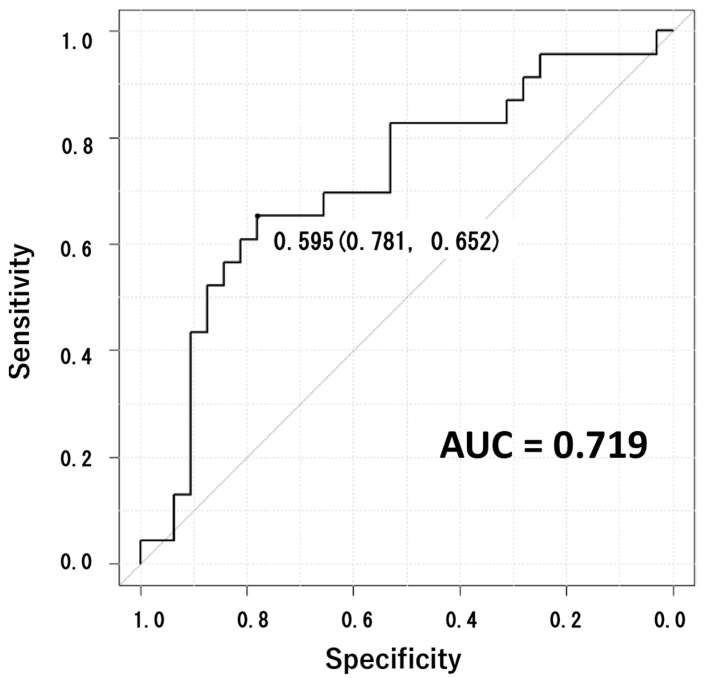
Applying model 1, the test was performed to distinguish each individual’s gait, whether pathological or not, resulting in an AUC of 0.719. The cut-off value (specificity and sensitivity) is shown near the curve.

**Figure 4 sensors-23-06217-f004:**
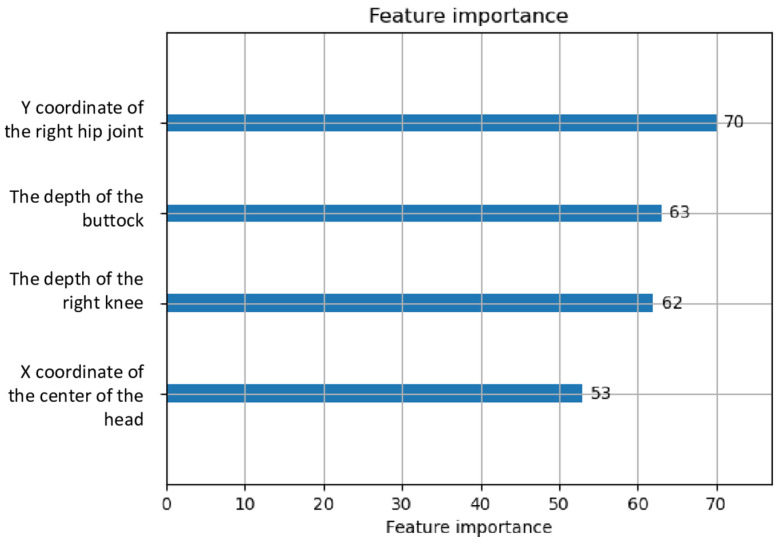
High feature importance scores in the discrimination of pathological gait.

**Table 1 sensors-23-06217-t001:** The number of participants in the datasets divided by the training, validation, and test.

	Training		Validation		Test		Total	
	Control	Disease	Control	Disease	Control	Disease	Control	Disease
Dataset 1	103	72	25	19	32	23	160	114
Dataset 2	102	73	26	18	32	23	160	114
Dataset 3	103	72	25	19	32	23	160	114
Dataset 4	104	71	24	20	32	23	160	114
Dataset 5	104	72	24	19	32	23	160	114

**Table 2 sensors-23-06217-t002:** The number, age, and sex proportion of the pathological gait group and the controls.

	Pathological Gait *n* = 114	Controls *n* = 160	*p*
Age (average ± SD *)	74.5 ± 7.8	72.9 ± 11.1	0.141
Sex (male/female)	52/62	91/69	0.076

* SD, standard deviation.

**Table 3 sensors-23-06217-t003:** Five datasets making up the results of the five models: the diagnosis of pathological gait evaluated by the cut-off, the area under the curve (AUC), sensitivity, specificity, and accuracy.

	Cut-off	AUC (95%CI)	Sensitivity	Specificity	Accuracy
Model 1	0.466	0.882 [0.875–0.890]	0.740	0.898	0.833
Model 2	0.415	0.932 [0.926–0.937]	0.852	0.878	0.868
Model 3	0.279	0.921 [0.915–0.926]	0.861	0.836	0.824
Model 4	0.273	0.820 [0.812–0.829]	0.798	0.723	0.716
Model 5	0.472	0.774 [0.764–0.784]	0.635	0.785	0.722

AUC: area under the receiver-operating characteristic curve; 95%CI: the parameter range within 95% confidence interval.

## Data Availability

Data generated or analyzed during the study are available from the corresponding author upon reasonable request.

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
