# Peer review of "Artificial Intelligence Distinguishes Pathological Gait: The Analysis of Markerless Motion Capture Gait Data Acquired by an iOS Application (TDPT-GT)"

_sensors, 2023, doi:10.3390/s23136217_

Round 1
Reviewer 1 Report
The paper needs more details in many sections. For example, the deep learning model -- what is the structure? what features are extracted? The cross-validation -- any test sets? Etc. Also the reference number is low, line 345 missing a citation.
Minor editing of English language required
Author Response
Thank you very much for your review. With the reviewers’ suggestions and advice, we could have had a better discussion of our results.
To reviewer1
Reviewer: The paper needs more details in many sections. For example, the deep learning model -- what is the structure?
>Thank you for your suggestions. I rewrote the paragraph starting line 91 in the methods section.
Reviewer: what features are extracted
>Thank you for your pointing out. Feature importance (FI) scores were shown in the result section (3.2.2). FI was generated during the tuning parameters, and it was meaningful for gait analysis. Therefore, we showed them in the result section.
Reviewer: The cross-validation -- any test sets? Etc
> Thank you for your pointing out. The expression was not clear in the previous manuscript. The cross-validation was undergone for 5 models in Table 1. And the method of the test was written in the next paragraph, shown below. To clarify, we now entitled the paragraph in methods.
Reviewer: Also the reference number is low, line 345 missing a citation.
> Thank you for your suggestion. I quoted references of more numbers. Now I paid attention to the citations.
Reviewer 2 Report
Artificial Intelligence Distinguishes the Pathological Gait: The Analysis of the Markerless Motion Capture Gait Data Acquired by the iOS Application (TDPT-GT) by Iseki et al.
Sensors-2433504-v1
This paper reports an interesting method of gathering and analyzing gait data to classify different pathological gait. Below are some suggestions that could make the paper stronger.
The first paragraph of the introduction provides the reader with a lot of background information. Please add references for this first paragraph so that reader can learn more about the topics if needed.
Only one reference for “much research” is listed in line 29. Please, include additional references or a review paper on the topic.
Use another word instead of “demented”, how about “individuals with mental problems”? (line 30)/
Verb tense is inconsistent in the later paragraphs of the introduction. Please, review.
Perhaps include some discussion on the use of video and machine learning for gait recognition in the paragraphs between lines 36 and 40. What has been reported up to date? Has it been successful? Is it mostly used systems with makers and sensors or with video systems such as the proposed here? Background discussion on the use of video capture and machine learning is needed in the introduction.
“Propose” instead of “purpose” in line 48?
The plots in Figure 2 are difficult to see. Numbers should be in a larger font. At least as large as the text of the narrative.
Reference missing in line 146.
Feature importance differences is a great outcome of this study. A more holistic observation rather than a focus marker seems more intuitive.
Compare your results to other published results. How does 0.702 accuracy compare to other methods?
The English language is mostly appropriate although it varies throughout the paper. For example, the verb tense is inconsistent in the later paragraphs of the introduction. Some of the sections are repetitive and some of the sentences are not properly formed, but the meaning is understood. I would recommend a detailed review of grammar.
Author Response
Thank you very much for your review. With the reviewers’ suggestions and advice, we could have had a better discussion of our results.
To reviewerï¼’
This paper reports an interesting method of gathering and analyzing gait data to classify different pathological gait. Below are some suggestions that could make the paper stronger.
The first paragraph of the introduction provides the reader with a lot of background information. Please add references for this first paragraph so that reader can learn more about the topics if needed.
Only one reference for “much research” is listed in line 29. Please, include additional references or a review paper on the topic.
> Thank you for your suggestion. With respect to your suggestion here and below, I added the phrases in the introduction part.
Use another word instead of “demented”, how about “individuals with mental problems”? (line 30)/
> Thank you for your suggestion. In medicine, we casually use the word “dementia” in papers, and “mental problems” mean beyond medical problems. Although, for readers in the journal, I rephrased it as “individuals with cognitive impairment.”
Verb tense is inconsistent in the later paragraphs of the introduction. Please, review.
> Thank you for your pointing. I corrected that.
Perhaps include some discussion on the use of video and machine learning for gait recognition in the paragraphs between lines 36 and 40. What has been reported up to date? Has it been successful? Is it mostly used systems with makers and sensors or with video systems such as the proposed here? Background discussion on the use of video capture and machine learning is needed in the introduction.
>Thank you for your suggestion. I added the description in the introduction part.
“Propose” instead of “purpose” in line 48?
> Thank you for your note. I rephrased that.
The plots in Figure 2 are difficult to see. Numbers should be in a larger font. At least as large as the text of the narrative.
>Thank you for your notice. We revised the Figure and noticed that it became better.
Reference missing in line 146.
> I am sorry for the elementary mistake. I corrected it.
Feature importance differences are a great outcome of this study. A more holistic observation rather than a focus marker seems more intuitive.
>Thank you for your suggestion. I added the discussion to the contents of feature importance.
Compare your results to other published results. How does 0.702 accuracy compare to other methods?
> Thank you for your suggestion. I added the discussion on that.
Reviewer 3 Report
The article “Artificial Intelligence Distinguishes the Pathological Gait: The Analysis of the Markerless Motion Capture Gait Data Acquired by the iOS Application (TDPT-GT)” demonstrates the use of cheap markerless tool and AI to identify pathological gait.
Point 1. The article requires improvement in teams of writing style in some sentences, grammatical proofreading, and referencing (for example line 144).
Point 2. It would be beneficial to provide more information in the methodology section regarding the specific low-pass filter used and the cut-off frequency. These details are important for other researchers to replicate the study and ensure methodological consistency.
Point 3. The article lacks a discussion how those findings can be utilised for the clinical use and what future studies required to achieve it.
Point 4. It would be valuable to provide recommendations for improving the models' sensitivity, specificity, and accuracy. Was 30 fps enough and could be some data missing due to low fps? 100 fps is widely used in clinical settings for gait analysis as well. Could be more iphones be beneficial to get better quality of data?
Point 5. There could be more discussion about potential challenges associated with AI analysis and providing suggestions for the future studies to address these challenges.
Author Response
Thank you very much for your review. With the reviewers’ suggestions and advice, we could have had a better discussion of our results.
To reviewer3
The article “Artificial Intelligence Distinguishes the Pathological Gait: The Analysis of the Markerless Motion Capture Gait Data Acquired by the iOS Application (TDPT-GT)” demonstrates the use of cheap markerless tool and AI to identify pathological gait.
Point 1. The article requires improvement in teams of writing style in some sentences, grammatical proofreading, and referencing (for example line 144).
> I am sorry for not having the best writing. The time limitation for this revision was short for final language editing (Just screened by native check). Line 144 was corrected. We are planning to commit to the MDPI editing service.
Point 2. It would be beneficial to provide more information in the methodology section regarding the specific low-pass filter used and the cut-off frequency. These details are important for other researchers to replicate the study and ensure methodological consistency.
>Thank you for your suggestions. I add the explanation about the filter in the methods part.
“These coordinates were processed by a low-pass filter, named 1 euro filter, setting the minimum cutoff frequency of 1.2, the cutoff slope of 0.001, and the cutoff frequency of derivate of 1.”
Point 3. The article lacks a discussion how those findings can be utilised for the clinical use and what future studies required to achieve it.
> Thank you for your suggestion. I added two paragraphs to the discussion part for future achievements.
Point 4. It would be valuable to provide recommendations for improving the models' sensitivity, specificity, and accuracy. Was 30 fps enough and could be some data missing due to low fps? 100 fps is widely used in clinical settings for gait analysis as well. Could be more iphones be beneficial to get better quality of data?
>Thank you for your view on gait analysis research. That was an important information; therefore, I added that into the discussion part.
Point 5. There could be more discussion about potential challenges associated with AI analysis and providing suggestions for the future studies to address these challenges.
> Thank you for your suggestion. I added two paragraphs to the discussion part for future studies.
Round 2
Reviewer 2 Report
The authors have addressed all of this reviewer's concerns.